# Fragility Curves of Existing RC Buildings Accounting for Bidirectional Ground Motion

Maria Zucconi [1], Marco Bovo [2,*] and Barbara Ferracuti [1]

1   Faculty of Engineering, Niccolò Cusano University, Via Don Carlo Gnocchi, 3, 00166 Rome, Italy;
    maria.zucconi@unicusano.it (M.Z.); barbara.ferracuti@unicusano.it (B.F.)
2   Department of Agricultural and Food Sciences, Alma Mater Studiorum Università di Bologna,
    Viale G. Fanin, 48, 40127 Bologna, Italy
*   Correspondence: marco.bovo@unibo.it

**Abstract:** In recent decades, the considerable number of worldwide earthquakes caused considerable damage and several building collapses, underlining the high vulnerability of the existing buildings designed without seismic provisions. In this regard, this work analyses the seismic performance of a reinforced concrete building designed without any seismic criteria, characterized by a seismically-stronger and a seismically-weaker direction, such as several existing reinforced concrete-framed structures designed for vertical load only. The case study building was modelled in OpenSees considering a non-linear three-dimensional model, also accounting for the contribution of joint panel deformability on the global behavior. Thirty bidirectional ground motions have been applied to the structure with the highest component alternatively directed along the two principal building directions. Time-history analyses have been performed for eight increasing hazard levels with the aim of evaluating the influence of bidirectional ground motion on structural response and estimating the seismic vulnerability of the building. The seismic performance of the structures are provided in terms of fragility curves for the two principal directions of the building and for different damage states defined according to the European Macroseismic Scale.

**Keywords:** RC structure; smooth bars; gravity load design; non-linear model; beam-column joint; fragility curves; damage thresholds

## 1. Introduction

Existing reinforced concrete (RC) structures built without seismic design criteria are widespread worldwide. For example, in Italy, a large percentage of the current building heritage was realized before modern building codes based on capacity design existed [1]. The current accepted seismic design philosophy establishes that a structure can face a seismic event without collapsing by reaching a high deformation level (typically in the inelastic field) but exhibiting ductile damage mechanisms, e.g., the formation of flexural plastic hinges on beams. The current capacity design method, developed firstly in academic contexts for the seismic design of buildings and nowadays included in many international codes [2–4], provides criteria for designing the structural elements and constructive details for a new generation of seismic-resistant buildings. On the other hand, the high seismic vulnerability of existing buildings leads to a great economic impact [5–8]. In this context, the seismic risk assessment of the existing building heritage and the risk reduction today represent crucial points, having the same importance as the design of new buildings [9–13].

It seems worth noting that numerical models for RC buildings have significantly been developed and applied over the past decades through complex numerical analyses and refined finite element (FE) models. In order to capture as best as possible the seismic behavior of a building and correctly assess its seismic performance, it is important to properly define the non-linear behavior of the various structural elements and consider the more effective seismic analyses to perform [14,15].

Today, especially in the academic field, non-linear time-history analyses that consider bidirectional ground motion applied to 3D models of buildings are becoming more and more widespread, even if these simulations provide outcomes not always easy to interpret [16]. Moreover, there are still open issues inherent both in structural elements modelling, e.g., the beam-column joint [17,18], the definition of ground motion input [19], the intensity measure choice [20], the record-to-record variability [21], and the epistemic uncertainty due to model [8,22] and the scaling of a bidirectional ground motion [23]. In fact, in a 3D structural model, the direction of application of the seismic input can greatly affect the assessment of the seismic performance [24,25], the evaluation of seismic damage to the structure [26] and, consequently, the estimate of economic losses [27,28]. However, the vulnerability of RC existing buildings depends on a great number of specific characteristics, such as quality and state of conservation of materials, the type of construction technologies adopted, the type of infills and their connection to the structural elements, the details of the nodal panels and the typology of floor diaphragms.

In the present paper, an existing RC building designed without any seismic criteria was selected as a representative case study and analysed under thirty bidirectional ground motion inputs for eight different hazard levels. The RC-framed building is characterized by a seismically-stronger direction and a seismically-weaker one. The behavior of the beam-column joints and the type of finite elements were carefully selected and defined in order to introduce in the FE model both ductile and brittle damage mechanisms characterizing existing buildings in the Mediterranean area. The non-linear dynamic analyses [29], performed for different seismic levels and driving the model until the structural collapse, aim to evaluate the influence of the beam-column joint deformability and of bidirectional ground motion on the structural performance expressed in terms of fragility curves for different damage states.

## 2. The Case Study

The building selected as the case study is representative of the structures built in the years 1960–1970 and designed without any seismic codes. The main criticalities of this class of structures, designed for vertical loads only, are:

- the lack or the total absence of capacity design provisions and details;
- realized with the philosophy of strong-beams and weak-columns;
- the presence of vertical loads resisting frames in only one of the main directions of the building;
- the lack of confinement effects in the panel joint regions;
- presence of low concrete and steel strength [18,30].

These structural deficiencies often promote brittle, local or global, failure mechanisms, for example, due to the attainment of the shear force capacity in the columns or brittle failure of the panel nodes at the lower floors [31,32]. In this regard, several works studied as the brittle mechanisms of bare RC frames influence the seismic vulnerability of the buildings [14,15,33]. For example, Jeon et al. [34] and Mohammad et al. [35] developed fragility curves of non-ductile RC frames for different building damage states, highlighting that the brittle mechanisms could influence the median value of fragility curves but also the scattering of the results, in particular at the severe damage limit states [18,22].

With reference to these aspects, the present paper considers the seismic analyses of an archetype building, representative of structures built before 1970s in the Mediterranean area and designed for vertical loads only, except wind. The plan view and the lateral view of the structure are shown in Figure 1. The building is a three-story RC-framed building designed according to the 1939 Italian building code provisions [36]. The structure is characterized by three bays with dimensions of 6.0 m each in the X-direction, and two bays with dimensions of 5.0 m in the Y-direction (see Figure 1). Beam and column cross-sections have the respective dimensions of 600 mm × 300 mm and 240 mm × 300 mm. The interstorey height is assumed to be equal to 3.0 m. The structure is located in Messina (Southern Italy) on class B soil [2]. In accordance with the building codes of 1939, a concrete

class C20/25 and a steel class (for longitudinal and transverse reinforcement bars) FeB32k have been assumed.

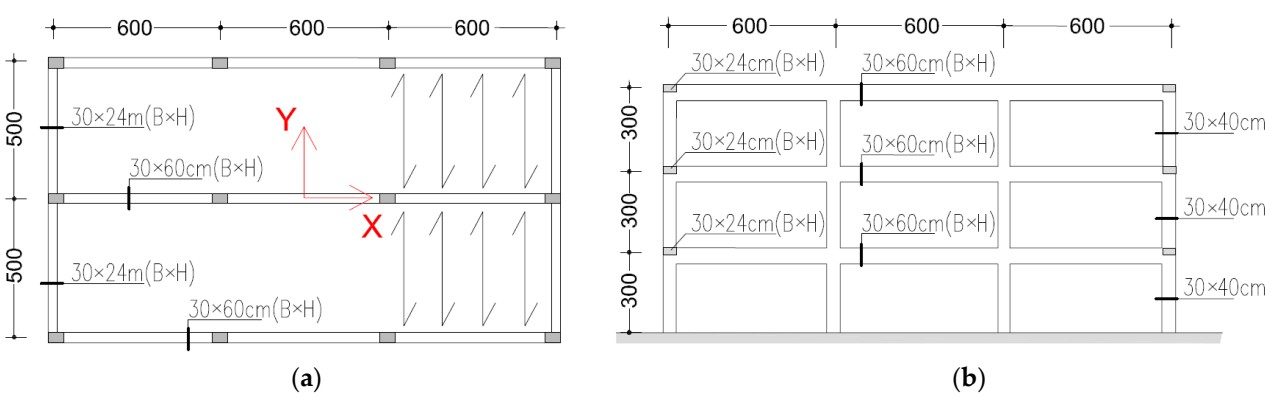

**Figure 1.** Geometry of the case study building. (**a**) Plan view; (**b**) Lateral view.

## 3. Ground Motion Selection and Scaling

In order to describe the performance of the three-dimensional case study building under an earthquake, the seismic input was defined by means of two acceleration time-history records applied along the two main horizontal directions of the models, i.e., X, Y (see axes in Figure 1).

The thirty pairs of time-histories were chosen and extracted from the Pacific Earthquake Engineering Research Center (PEER) strong motion database [37] considering: a magnitude moment Mw ranging from 6.5 to 8.0; a distance from the hypocentre ranging from 6 to 50 km; a horizontal peak ground acceleration (PGA) varying from 0.07 g to 0.48 g recorded on site classes B or C according to Eurocode 8 [3]. The $PGA_1/PGA_2$ ratio varies between 0.5 and 2.0, as shown in Table 1. Figure 2 shows the horizontal geometric mean (geo-mean) of elastic acceleration response spectra for a 5% soil damping ratio (grey lines) and mean of the 30 response spectra (red line). Each record pair was applied twice to the structure by rotating 90°. Therefore, the largest spectral acceleration was applied first in the X direction and then in the Y direction.

**Table 1.** Time history records adopted for the analysis (* Moment magnitude; ** Closest distance to fault rupture; *** Fault mechanism, where R: reverse; SS: strike-slip; RO: reverse-oblique).

| No. | Event Name | Station | M * | R ** [km] | Mech. *** | $PGA_1$ [g] | $PGA_2$ [g] |
|---|---|---|---|---|---|---|---|
| 1 | San Fernando, 1971 | LA-Hollywood Stor FF | 6.61 | 22.77 | R | 0.225 | 0.163 |
| 2 | Imperial Valley-06, 1979 | Parachute Test Site | 6.53 | 12.69 | SS | 0.113 | 0.206 |
| 3 | Superstition Hills-02, 1987 | Brawley Airport | 6.54 | 17.03 | SS | 0.131 | 0.111 |
| 4 | Superstition Hills-02, 1987 | Poe Road (temp) | 6.54 | 11.16 | SS | 0.475 | 0.286 |
| 5 | Spitak_Armenia, 1988 | Gukasian | 6.77 | 23.99 | RO | 0.200 | 0.174 |
| 6 | Loma Prieta, 1989 | Coyote Lake Dam-SW | 6.93 | 19.97 | RO | 0.132 | 0.280 |
| 7 | Loma Prieta, 1989 | Fremont—Emerson Court | 6.93 | 39.66 | RO | 0.192 | 0.099 |
| 8 | Landers, 1992 | Mission Creek Fault | 7.28 | 26.96 | SS | 0.097 | 0.132 |
| 9 | Northridge-01, 1994 | LA—Pico & Sentous | 6.69 | 27.82 | R | 0.103 | 0.186 |
| 10 | Northridge-01, 1994 | LA—S. Vermont Ave | 6.69 | 27.89 | R | 0.137 | 0.068 |
| 11 | Northridge-01, 1994 | LA—Temple & Hope | 6.69 | 28.82 | R | 0.124 | 0.165 |
| 12 | Kobe_Japan, 1995 | Abeno | 6.90 | 24.85 | SS | 0.149 | 0.231 |
| 13 | Denali_Alaska, 2002 | Carlo (temp) | 7.90 | 49.94 | SS | 0.081 | 0.084 |
| 14 | San Simeon_CA, 2003 | Cambria-Hwy1 Caltrans Bridge | 6.52 | 6.97 | R | 0.179 | 0.126 |
| 15 | Niigata_Japan, 2004 | FKS028 | 6.63 | 30.11 | R | 0.135 | 0.170 |
| 16 | Niigata_Japan, 2004 | NIG023 | 6.63 | 25.33 | R | 0.405 | 0.248 |
| 17 | Chuetsu-oki_Japan, 2007 | Nadachiku Joetsu City | 6.80 | 35.79 | R | 0.119 | 0.155 |
| 18 | Chuetsu-oki_Japan, 2007 | Tokamachi Chitosecho | 6.80 | 25.35 | R | 0.201 | 0.251 |
| 19 | Chuetsu-oki_Japan, 2007 | Kawaguchi | 6.80 | 23.63 | R | 0.147 | 0.147 |
| 20 | Chuetsu-oki_Japan, 2007 | NIG022 | 6.80 | 37.79 | R | 0.155 | 0.126 |

**Table 1.** *Cont.*

| No. | Event Name | Station | M * | R ** [km] | Mech. *** | PGA₁ [g] | PGA₂ [g] |
|---|---|---|---|---|---|---|---|
| 21 | Iwate_Japan, 2008 | IWT010 | 6.90 | 16.26 | R | 0.226 | 0.289 |
| 22 | Iwate_Japan, 2008 | Kami_ Miyagi Miyazaki City | 6.90 | 25.15 | R | 0.117 | 0.156 |
| 23 | Iwate_Japan, 2008 | Iwadeyama | 6.90 | 20.77 | R | 0.269 | 0.354 |
| 24 | Iwate_Japan, 2008 | Oomagari Hanazono-cho_Daisen | 6.90 | 46.32 | R | 0.093 | 0.127 |
| 25 | Iwate_Japan, 2008 | Mizusawaku Interior O ganecho | 6.90 | 7.82 | R | 0.361 | 0.257 |
| 26 | Darfield_New Zealand, 2010 | DFHS | 7.00 | 11.86 | SS | 0.275 | 0.333 |
| 27 | Darfield_New Zealand, 2010 | DORC | 7.00 | 29.96 | SS | 0.070 | 0.084 |
| 28 | Darfield_New Zealand, 2010 | OXZ | 7.00 | 30.63 | SS | 0.119 | 0.105 |
| 29 | Darfield_New Zealand, 2010 | RKAC | 7.00 | 13.37 | SS | 0.167 | 0.191 |
| 30 | Cucapah_Mexico, 2010 | El Centro Array #4 | 7.20 | 35.08 | SS | 0.238 | 0.310 |

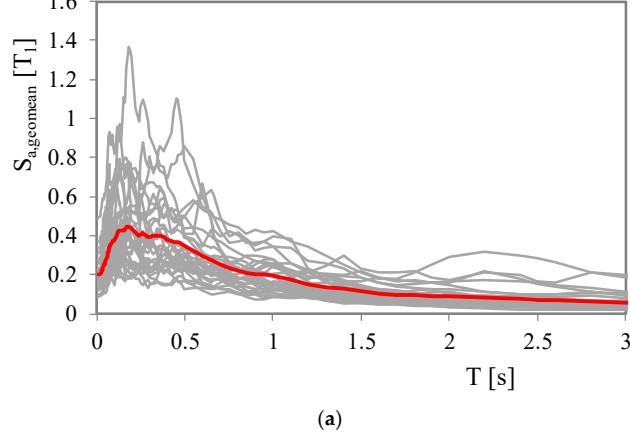

(**a**)

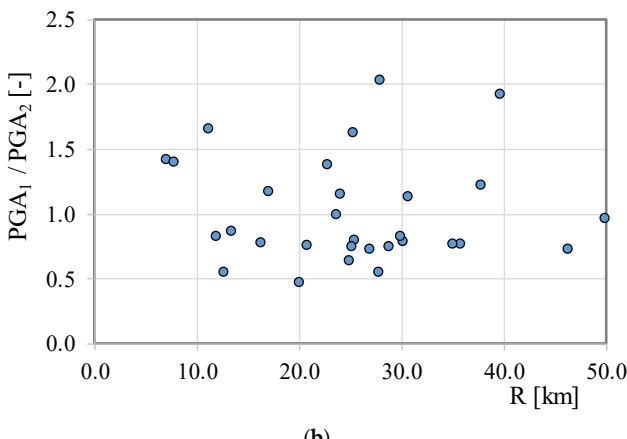

(**b**)

**Figure 2.** Main characteristics of the adopted accelerograms. (**a**) Elastic acceleration response spectra; (**b**) $PGA_1/PGA_2$ Vs. R (distance to fault).

The structural seismic performances were assessed by means of time-history analyses performed for eight different hazard levels (HLs) characterized by different return periods. The selected HLs, expressed as the probability of exceedance in a period of 50 years, were: 2%, 5%, 7%, 10%, 20%, 30%, 63% and 81%.

## 4. Finite Element Modelling

The seismic analyses were performed on a detailed finite element model of the building. Numerical modelling and dynamic analyses were performed by the OpenSees software [38]. A non-linear model has been considered in order to capture, for the increasing seismic intensity, the degradation mechanisms of the building until the building collapse.

The 3D bare frame model of the structure was realized by employing 1D elastic finite elements for beams and columns, and the non-linear behavior was accounted for by means of flexural and shear springs introduced with zero-length elements, as shown in Figure 3. At the two extremities of column and beam elements, two zero-length elements were added. A first zero-length element with rigid-plastic behavior was used to introduce the element's shear collapse mechanism [39]. The points of the curve were defined according to [3], evaluating the shear strength capacity starting from the sectional analysis as a function of the adopted shear reinforcement. Their inelastic behavior was modelled with the Hysteretic material in OpenSees, as shown in Figure 4a. A second zero-length element was added at the extremity of beams and columns, in series with the shear zero-length element, in order to model the inelastic flexural behavior. The trilinear moment-rotation plastic hinges were modelled with Hysteretic material. The values were obtained starting from the properties of the cross-sections of the different elements and material properties of concrete and steel for reinforcement bars. The typical backbone curve adopted for modelling the flexural plastic hinge is shown in Figure 4b. The backbone curve is defined by means of the values of moment and rotation at the yielding point ($M_y$, $\Theta_y$) and at the failure point, i.e., the

ultimate values, $(M_u, \Theta_u)$. The main points of the backbone curves of the plastic hinges were calculated as recommended in [2] for existing buildings, with the bending moment in the columns calculated by taking into account the axial compression forces due to the presence of gravitational loading.

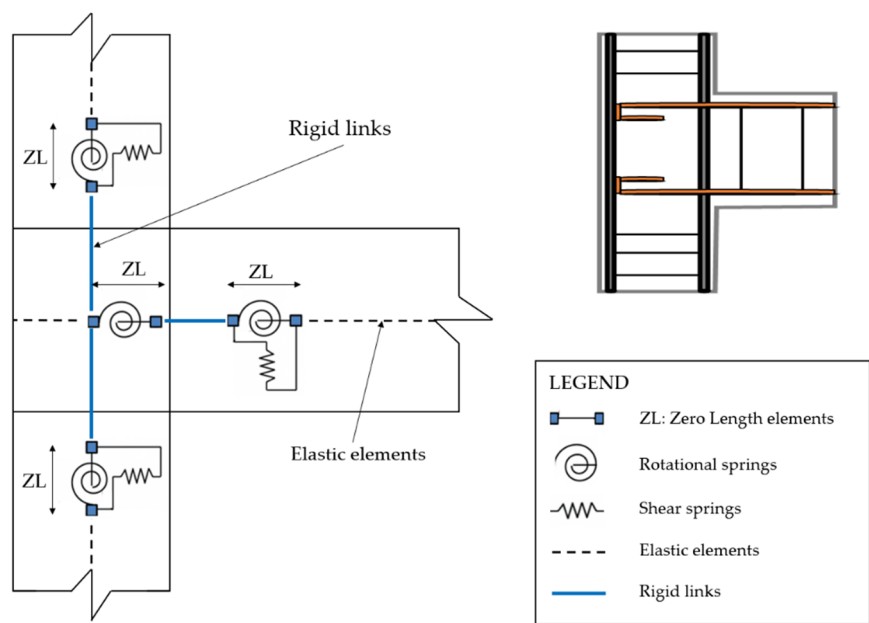

**Figure 3.** Details of the FE model of the beam-column joint.

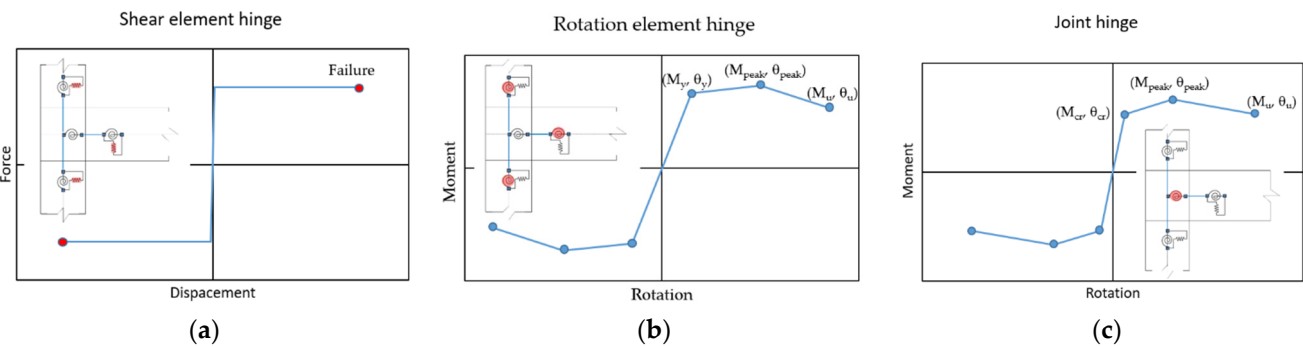

**Figure 4.** Non-linear behavior adopted in the FE models. (**a**) Zero-length elements at the extremities of beams and columns; (**b**) Plastic hinges at the extremities of beams and columns; (**c**) Zero-length elements simulating the non-linear behavior of the joint panels.

A further zero-length element was added at the extremity of the columns, in series with the shear zero-length element, in order to model the joint flexibility and introduce the possible joint collapse mechanism [17]. The non-linear behavior of the joint was introduced in the model as trilinear behavior calibrated as discussed in [40] on the basis of experimental results in [41,42]. The moment-rotation law, displayed in Figure 4c, is characterized by different capacity values for each floor, as a function of the axial load, and for the exterior or internal panel joints. The aged properties of the materials adopted in the model for the definition of the different plastic hinges are the following: $f_c$ = 24 MPa and $E_c$ = 30.94 GPa (respective compression stress strength and Young modulus) for concrete, whereas the values $f_y$ =375 MPa and $E_s$ = 210 GPa (respective yielding stress and Young modulus) have been assumed for the steel of the reinforcement bars. At the ground level, the columns were fully clamped, neglecting the soil-structure interaction. Masses corresponding to structural dead loads, non-structural dead loads and live loads were considered as equivalent distributed masses on the beams.

The collapse condition has been defined in the time-history analyses as the attainment of one of the following sub-conditions:

- the ultimate rotation for a column;
- the ultimate rotation for a beam;
- the displacement capacity in one of the shear sliding hinges;
- an interstorey drift ratio equal to 5%.

## 5. Structural Analysis Results

For each HL, the probabilistic seismic response is evaluated in terms of engineering demand parameters (EDP) distributions, evaluated as a function of the peak interstorey drift ratio, IDR, recorded in both X and Y directions. The structure has elastic vibrating periods equal to $T_{1X}$ = 0.40 s and $T_{1Y}$ = 0.75 s, respectively, in the X and Y directions.

Figure 5 compares the numerical model results for the two directions X and Y, in terms of median peak IDR profiles at 63%, 30% and 10% in 50 years of HLs, revealing the different evolution of floor damage in the two considered directions. The IDR progression is larger for the Y direction, showing a higher vulnerability than the X direction. This behavior is already observable at lower HLs (Figure 5a) and became more relevant for higher HLs (Figure 5b,c).

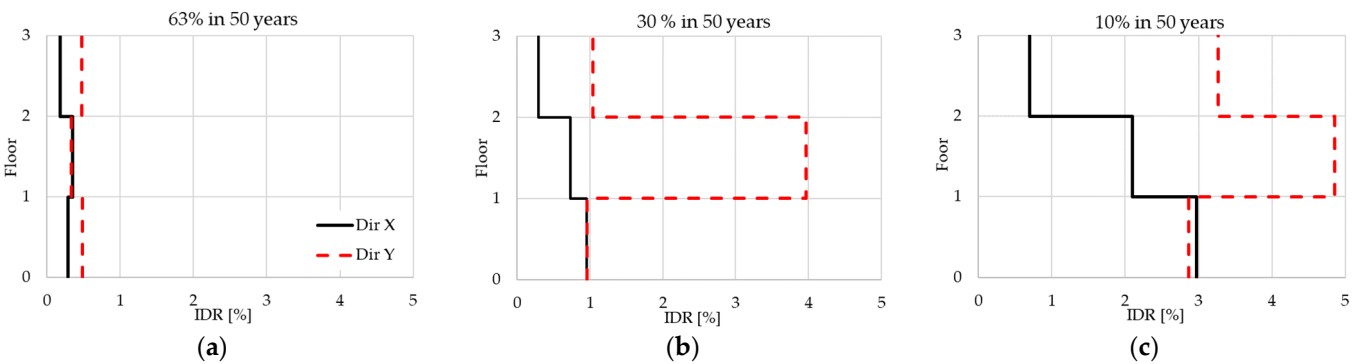

**Figure 5.** Median values of IDR for the time-history analyses at (**a**) 63%, (**b**) 30% and (**c**) 10% in 50 years of HLs (solid line X direction, dashed line Y direction).

At 63%, in 50 years of HLs, the building shows an elastic behavior in the X direction, whereas in the Y direction only, some beams' plastic hinge is activated. At 30%, in 50 years of HLs, the building shows a plastic behavior in both directions: in the X direction, some panel joints reach the first cracking condition, whereas beams and columns remain in the elastic range; contrarily, in the Y direction, the buildings collapse occurs; in fact, several structural elements (beams, columns and joints) enter in their plastic range and reach their ultimate state condition. In the X direction, only at higher HLs, the activation of columns plastic hinges is registered, followed by the beams. The shear plastic hinge is never activated in both directions, and the flexural behavior is always prevalent for all structural elements. In Figure 5b–c, it is possible to note that the maximum IDR is concentrated on the second floor for the Y direction, where a soft story mechanism occurred, and on the first floor for the X direction. At 10%, in 50 years of HLs, the median IDR values are almost 5% in Y and 3% in X, and building collapse occurred in both directions.

Figure 6 shows the structural analyses results in terms of IDA (Incremental Dynamic Analyses) curves representing the maximum IDR evaluated over the whole building vs the values of the geometric-mean spectra acceleration Sa ($T_1$): reporting the trend of 50% (solid red line), 16%, and 84% percentiles (dashed red lines) for drift distribution and the results of the single time-history (blue circles).

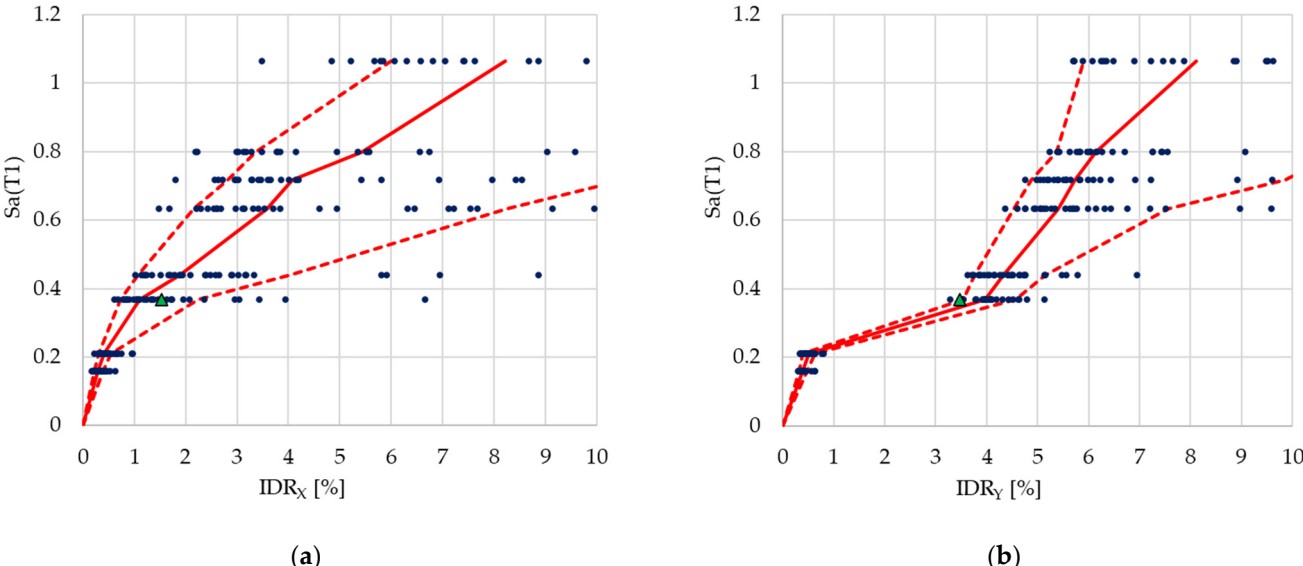

**Figure 6.** Peak interstorey drift ratio (IDR) vs Sa ($T_1$). (**a**) Case I with IDR measured in X direction; (**b**) Case II with IDR measured in Y direction. The green triangles indicate the results of the first time-history of Table 1.

IDA curves confirm the different behavior of the buildings in the two directions. In the first two analysed HLs, (81% and 63% in 50 years), the building shows an elastic behavior in both directions, and the initial stiffness is comparable. Then, from 63% to 30% in 50 years of HLs, a significant decrease in stiffness can be observed from IDA curves in the Y direction (Figure 6b), suggesting a collapse condition for the building. Conversely, the structure retains a significant stiffness in the X direction, gradually reducing.

Then, the structural results of the first time-history of Table 1 at 30% in 50 years of HLs, which is identified in Figure 7 with a green triangle, will be discussed in detail in order to underline the importance of the joint panel modelling in the seismic performance of buildings and, consequently, on the fragility curves definition. In particular, the rotation contribution of each structural element was registered during the time-history analysis. With reference to an exterior beam-column joint sub-system between the first and the second floor, the rotation contributions of the joint panel, upper column and lower column in the X direction are shown in Figure 7 for two selected analysis time steps: the attainment of the maximum interstorey drift ratio in Figure 7a, and the attainment of the maximum rotation at the investigated node in Figure 7b. The beam rotation is not reported because its contribution is negligible if compared to those of columns and joints. In fact, due to the absence of the capacity design criterion, the building shows a strong beam-weak column mechanism with beams characterized by almost zero rotations. In both Figures, it is possible to note that the rotation contribution is relevant, and it grows from 40% to more than 70%. At the maximum interstory drift, the upper columns and the joint panel give the same rotation contribution equal to 40% and the lower column contribution is 20% (Figure 7a). At the maximum rotation of joint panel (Figure 7b), the lower column contribution is negligible compared to the upper column rotations and joint one, equal to 27% and 71%, respectively.

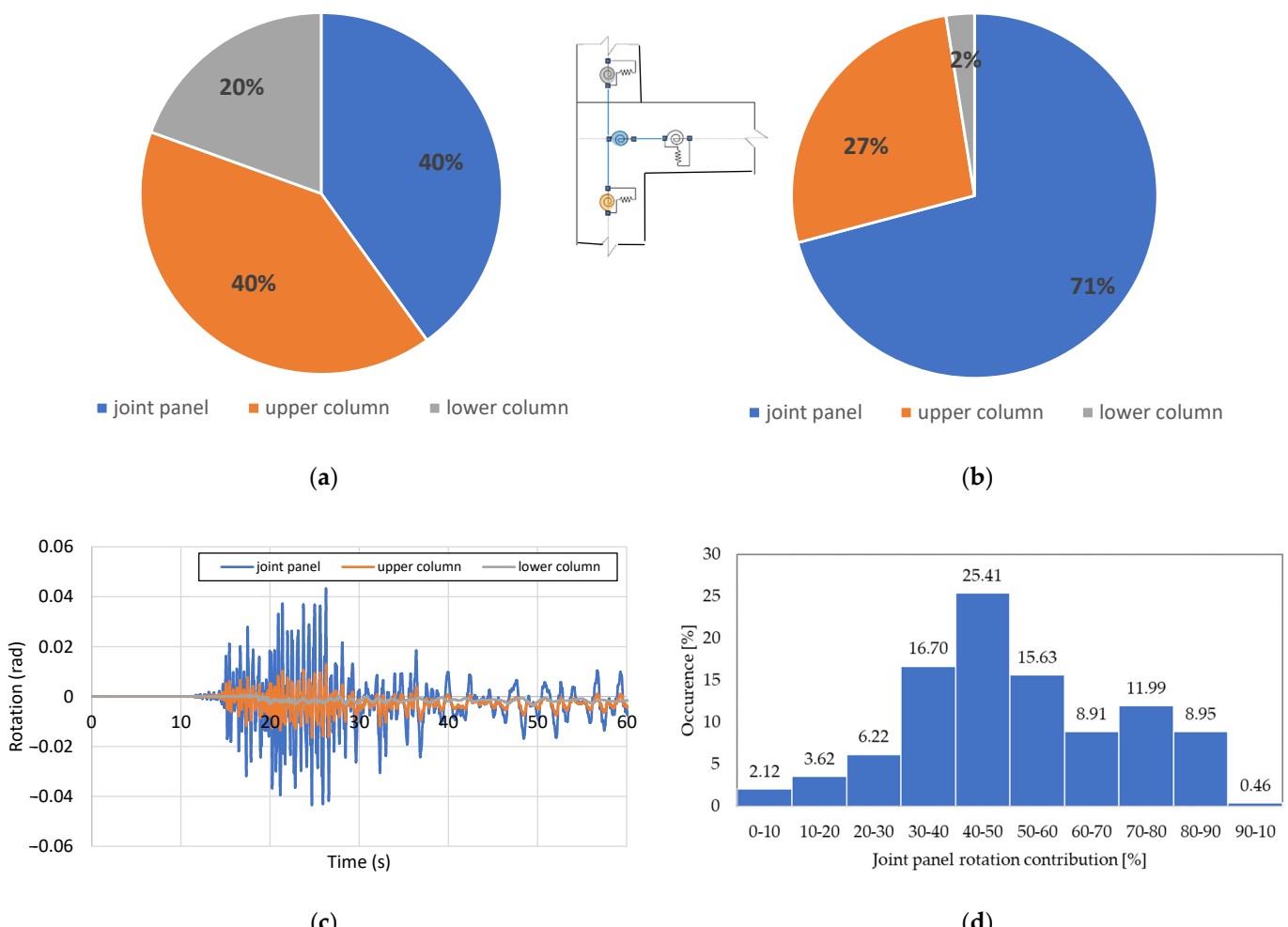

**Figure 7.** Deformability contribution of joint panel, upper column and lower column to the total rotation on an external node of the first floor during the first time-history analyses of the intensity level at 30% in 50 years HL. (**a**) Time-step corresponding to the attainment of the maximum interstorey drift ratio (instant 22.23 s of the time history); (**b**) Time-step corresponding to the attainment of the maximum rotation at the investigated joint panel (instant 27.35 s of the time history); (**c**) Example of rotation recorded for the joint panel, upper column and lower column; (**d**) Histograms diagram showing the distribution of the percentage values of the joint panel contribution during the time-history analyses.

The rotation contribution of the structural members converging in the beam-column joint sub-system varies during the dynamic structural analysis, as shown in Figure 7c, where the rotation recorded for the selected joint panel, upper column and lower column is reported for the first 60 s (out of a total of 120 s) of time-history analysis. After the first 60 s, the rotation values are less significant and move toward zero. As can be noted, the results shown in Figure 7a,b are representative of the rotation contribution of the structural elements with respect to the total deformation demand. In fact, the joint panel and the upper column provide the more relevant rotation contribution in all analyses. This result is also compatible with the IDR profiles presented in Figure 5, where higher displacement occurs on the second floor in the Y direction.

Finally, in Figure 7d, the histogram diagram displays the percentage values distribution of the joint panel rotation contribution during the time-history analysis. It is possible to note that in more than 71% of the cases, the joint rotation contribution is greater than 40%, and in more than 45% of the cases, the joint contribution is greater than 50%. The median rotation contributions are approximately equal to 48%, 36% and 17% for the joint

panel, the upper column and the lower column, respectively. Therefore, in more than 67% of the total cases, the upper column gives a contribution of less than 40%, and in 68% of the total cases, the lower column contribution is 20%.

Furthermore, by the analysis of the typical sequence of the plastic hinges activation (see Figure 8), it emerges that the first non-linearities in the building arise at the level of vertical columns. In the figure, the first number is the order of activation during the time-history, whereas the second is the activation time step (in seconds). The blue circles indicate flexural hinge activation on columns, green circles indicate flexural hinge activation on beams and red circles indicate rotational hinge activation on the joint panel. Then, in the Y direction, the direction weakens in terms of both strength and stiffness, and the activation of the flexural plastic hinges of the beams anticipates the activation of the rotational hinges of the joint panels. The opposite occurs in the X direction where the joint panels enter in the non-linear range before the activation of the flexural hinges of the beams. This confirms the importance of modelling nodal joints flexibility in order to obtain a reliable evaluation of the non-linear behavior of existing RC structures not designed with modern criteria and details.

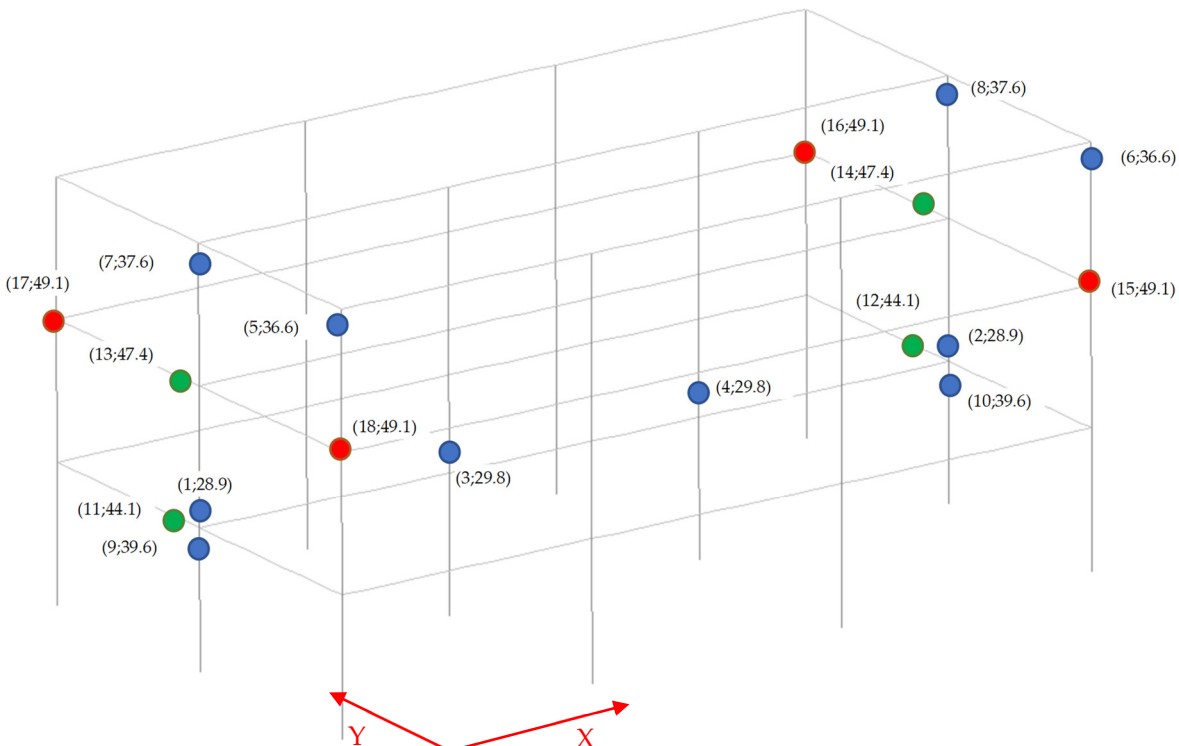

**Figure 8.** Activation sequence of the different non-linear behaviours attributed to the different zero-length elements.

## 6. Damage States and Fragility Curves

The structural analysis results, expressed in terms of IDR, were employed as EDP for building vulnerability assessment through fragility curves.

Following other literature works [43,44], the lognormal distribution was selected to fit the numerical points in order to derive the fragility curves, according to the following equation:

$$P[DS \geq DS_i | PGA] = \Phi\left(\frac{\ln(PGA) - \mu}{\beta}\right) \qquad (1)$$

where $P[DS \geq DS_i | PGA]$ is the probability of reaching or exceeding a specific damage state $DS_i$ given a $PGA$ value; $\Phi(\cdot)$ is the standard normal cumulative distribution function; $\mu$ is the logarithmic mean and $\beta$ is the logarithmic standard deviation.

Similar to other literature work, the parameters $\mu$ and $\beta$ were evaluated by means of the maximum likelihood estimation method, assuming that the numerical points follow the binomial distribution (e.g., [12,43,45]). Moreover, to avoid the intersection of curves, the parameter $\beta$ was constrained to be equal or increasing with higher damage states $DS_i$ [46,47].

Damage states were defined according to the EMS-98 scale from DS1 to DS5. In order to define the numerical thresholds for the selected EDP, two different criteria were adopted: HAZUS-MH MR5 (2009) [48] and Borzi et al. 2008 [49] for which the association with the EMS-98 damage states is suggested in this work, as reported in Table 2. In particular, the HAZUS-MH MR5 manual proposes different values of damage states expressed in terms of IDR, whose values are defined on the basis of the observed damage and of expert opinions. Furthermore, Borzi et al. 2008 define three damage states based on the rotational capacity of a single RC column. In particular, light damage state corresponds to the column yielding rotation $\theta_y$, significant damage state corresponds to $\frac{3}{4}$ of the rotational capacity $\theta_u$ and collapse limit condition rotation corresponds to the attainment of the column rotational capacity $\theta_u$. Then, three-chord rotation limit values according to the geometrical and mechanical characteristics of the vertical seismic-resistant structural elements can be calculated. Table 2 indicates the calculated values for the case study and their association to the damage state of EMS-98. Therefore, in order to use the thresholds adopted by Borzi et al. in 2008, the IDR is assumed to be comparable to the required columns' chord rotation neglecting the contribution of joints and beams.

**Table 2.** IDR damage states thresholds (%) as a function of the different criteria considered.

| Reference | DS1 | DS2 | DS3–DS4 | DS5 |
|---|---|---|---|---|
| HAZUS MH-MR5 TM (2009) | 0.4% | 0.6% | 1.6% | 4.0% |
| Chord Rotation CR [49] | - | 1.1% ($\theta_y$) | 2.5% (3/4 $\theta_u$) | 3.3% ($\theta_u$) |

Figure 9 depicts the calibrated damage fragility curves for the X and Y directions, considering the damage thresholds according to HAZUS and CR criterion.

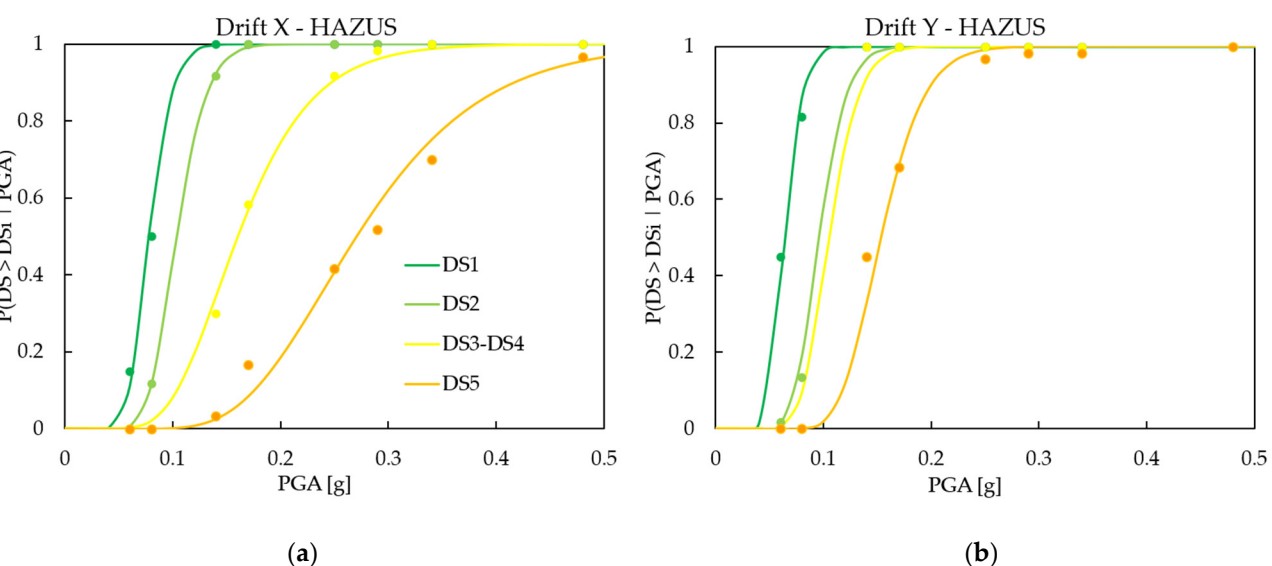

(**a**) (**b**)

**Figure 9.** *Cont.*

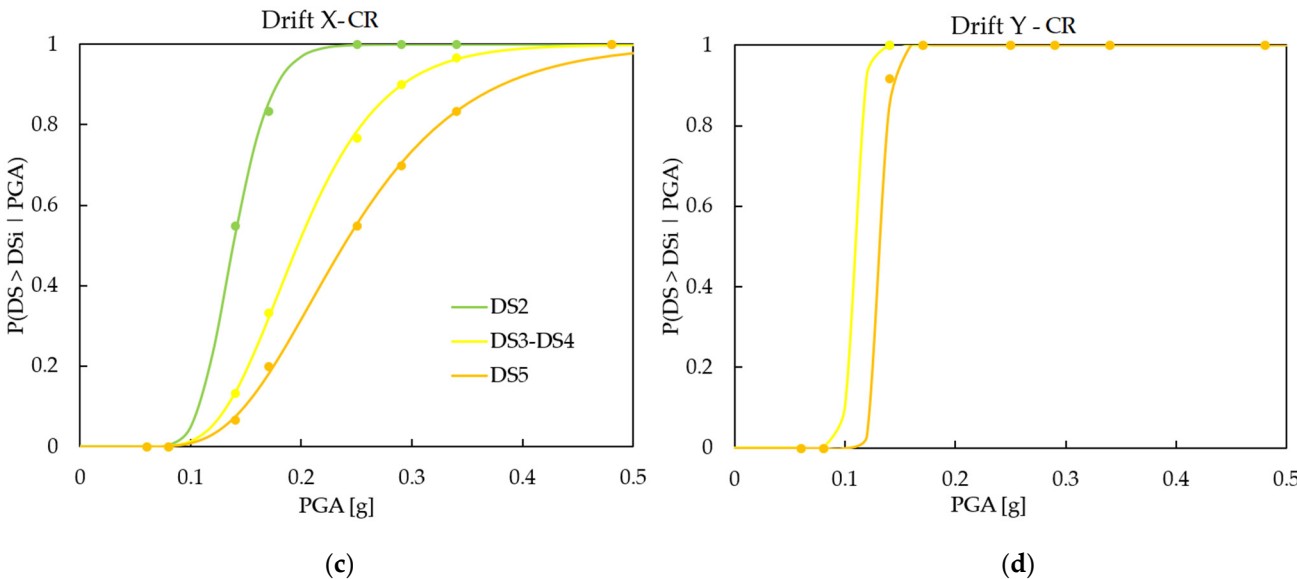

**Figure 9.** Fragility curves with damage thresholds defined according to HAZUS (**a**,**b**) and CR criterion (**c**,**d**) for X (**a**,**c**) and Y (**b**,**d**) directions.

The results show that the three-dimensional modelling of the structure and the application of bi-directional ground motion are relevant aspects to consider for the calibration of the fragility curves. In particular, the outcomes reveal how the vulnerability of the structure is different in the two directions, highlighting a greater vulnerability in Y direction. Therefore, in the Y direction, the curves are characterized by a higher slope, demonstrating a higher fragility of the structure for both damage thresholds defined according to HAZUS and CR criterion. This result is consistent with the IDA curves shown in Figure 6b, where it is possible to notice an abrupt collapse of the structure at 30% in 50 years of HLs. In the X direction, the fragility curves indicate a lower vulnerability of the structure, coherent with the structural analysis results (see Figure 6a), where the IDA curves exhibit a progressive collapse of the building. This structural behavior is also confirmed by the results of Table 3 and Figure 10, in which it can be observed that the median values of the fragility curves are always lower in the Y direction, thus showing a higher vulnerability of the structure for both the thresholds employed. In Figure 9d, the fragility curves for damage states DS2 and DS3-DS4 are overlapped because for PGA ≥ 0.14 g, for all time histories, the higher damage state threshold is reached.

**Table 3.** Parameters of lognormal fragility curves for X and Y direction and for the adopted IDR damage states thresholds.

| | HAZUS | | | | CR Criterion | | | |
|---|---|---|---|---|---|---|---|---|
| Damage State | Drift X | | Drift Y | | Drift X | | Drift Y | |
| | Exp($\mu$) [g] | $\beta$ | Exp($\mu$) [g] | $\beta$ | Exp($\mu$) [g] | $\beta$ | Exp($\mu$) [g] | $\beta$ |
| DS1 | 0.078 | 0.213 | 0.064 | 0.207 | - | - | - | - |
| DS2 | 0.103 | 0.213 | 0.096 | 0.207 | 0.138 | 0.196 | 0.109 | 0.067 |
| DS3–DS4 | 0.160 | 0.338 | 0.105 | 0.207 | 0.197 | 0.306 | 0.109 | 0.067 |
| DS5 | 0.270 | 0.338 | 0.153 | 0.207 | 0.238 | 0.368 | 0.133 | 0.054 |

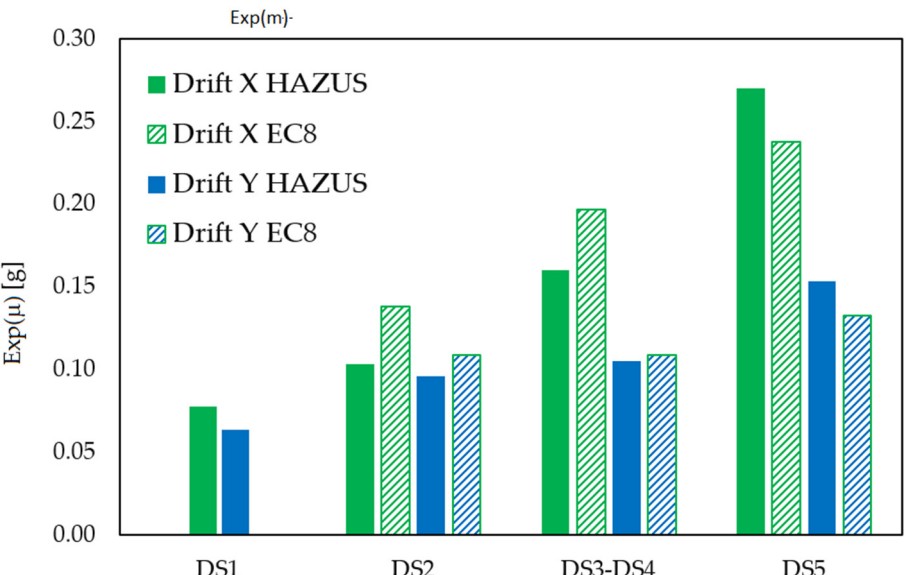

**Figure 10.** Median $\exp(\mu)$ of fragility curves for X and Y direction and for the adopted IDR damage state thresholds.

Finally, with regard to the damage states thresholds, it can be noted in Figure 9 that a higher vulnerability is registered using the damage thresholds proposed by HAZUS for DS2 and DS3-DS4, whereas for DS5, CR criterion is more restrictive. This result is consistent with the threshold values reported in Table 2, which turn out to be higher in the case of CR criterion for the first two DS, whereas for DS5, the IDR value is lower in CR criterion than in HAZUS (3.3% against 4%). This trend is clearly visible in Table 3 and Figure 10.

It is worth remembering that the threshold values in terms of IDR for CR criterion are calculated as a function of the yield and ultimate rotation of the structural members; therefore, buildings with structural elements characterized by different mechanical and geometrical properties with respect to the case study could lead to different results.

Concerning the standard deviation, from the values reported in Table 3, it can be observed that the values of dispersions are always lower in the Y direction than in the X direction for both threshold criteria. In contrast, in the X direction, the dispersions obtained by HAZUS and CR criteria are comparable, whereas in the Y direction, they are lower for CR criterion.

## 7. Conclusions

This work focuses on evaluating the seismic performance of an RC frame building that is not built according to seismic design standards. The paper's main aim is to assess the seismic vulnerability of the structure through the calibration of numerical fragility curves. Particular consideration was given to the definition of the structural modelling, the characterization of the seismic input and the identification of damage states thresholds for the calibration of the fragility curves.

The case study building consists of a three-dimensional model implemented in OpenSees. The non-linear FE model accounts for the shear and flexural behavior of beams and columns for which suitable constitutive laws were defined. Moreover, the joint panel deformability was carried out by introducing a moment-rotation constitutive relationship to simulate his non-linear behavior. For a specific NLTH, the contribution of exterior beam-column joint on the interstorey drift was evaluated, showing a mean value of 48% with respect to the deformation provided by the sub-system composed of beam, column and joint.

The seismic input was defined through bidirectional ground motions by selecting thirty pairs of acceleration time-histories from the Pacific Earthquake Engineering Research Center (PEER) database. The higher acceleration of selected records was applied to the

building along with the two main horizontal directions of the models. The accelerograms pairs were rotated by 90° degrees in order to account for the response variability due to the directionality of ground motion. Time-history analyses were performed for eight different HLs. The structural results on peak inter-story drift ratio, allowed for analysing the seismic performance of the building and to derive fragility curves for damage states defined according to the EMS-98. The main outcome can be synthesized as follows:

- The non-linear three-dimensional model together with the bi-directional ground motion allowed for highlighting a different seismic behavior of the structure in the two main directions (X and Y), revealing a higher vulnerability in the Y direction with respect to X. The RC bare frame in the Y direction reaches collapse at 30% in 50 years of HLs, whereas in the X direction, at the same HLs, the first cracking condition in some external joints is achieved.
- The non-linear behaviours attributed to structural elements account for shear and flexural behavior of beams and columns, and the moment-rotation relationship attributed to the joint panel allowed us to underline the different activation sequences in the two main directions of the building. In particular, it is possible to identify a structural behavior governed by the bending failure of beams and columns in the Y direction and a behavior controlled by the shear failure of the joint in the Y direction.
- The damage thresholds are defined following two criteria: the first one proposed by HAZUS-MH MR5 (2009), with given values of IDR depending on the different classes of RC frame buildings; the second one [46] is a local criterion in which the thresholds are defined on the basis of the ultimate and yielding rotation of columns. The adopted criteria significantly affect the fragility curves shape and the parameters values of the lognormal distribution adopted to fit the numerical points. In particular, for the damage state DS2–DS3 and DS4, the HAZUS criterion is more conservative, leading to a higher vulnerability characterized by lower median values of the parameter distribution. Contrarily, for DS5, the local criterion is more restrictive.
- Consistent with the numerical results, the fragility curves show a higher vulnerability for the Y direction of the building, with lognormal distribution median values lower than in X direction for both damage thresholds criteria. Nevertheless, with the local criterion, the slope of the functions is more marked, showing that the overcoming of the damage state occurs nearly always for the various time-history analyses and the different HLs.

**Author Contributions:** Conceptualization, M.Z., M.B. and B.F.; methodology, M.Z., M.B. and B.F.; software, M.Z. and M.B.; validation M.Z., M.B. and B.F.; formal analysis, M.B. and M.Z.; investigation, M.Z., M.B. and B.F.; resources, B.F.; data curation, M.Z. and M.B.; writing—original draft preparation, M.Z. and M.B.; writing—review and editing, M.Z., M.B. and B.F.; visualization, B.F.; supervision, B.F.; project administration, B.F.; funding acquisition, B.F. All authors have read and agreed to the published version of the manuscript.

**Funding:** This work was developed under the financial support of the Italian Civil Protection Department within the DPC-ReLUIS 2022–2024 research project, which is gratefully acknowledged.

**Data Availability Statement:** Data sharing is not applicable to this article.

**Acknowledgments:** The authors wish to acknowledge the financial support received by the Italian Department of Civil Protection (ReLUIS 2022–2024 Grant—Inventory of existing structural and building types- CARTIS).

**Conflicts of Interest:** The authors declare no conflict of interest.

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
