# Peer review of "Fragility Curves of Existing RC Buildings Accounting for Bidirectional Ground Motion"

_buildings, doi:10.3390/buildings12070872_

Round 1

Reviewer 1 Report

The authors made a great job in analyze the seismic performance of a RC building constructed in Mediterranean area based on non-seismic design codes. Opensees is adopted to handle the FEA and fragility curves for different damage states are provided. The reviewer has following questions for the authors:

How the foundation and soil-structure interaction is modeled/considered in the FE model?

Please explain detailed procedures to acquire/define the values used in zero-length elements simulating nonlinear behavior, as in Fig 3 a,b and c, including My,Mpeak,Mcr, etc..

A figure showing whole FE model, indicated with detailed location of nonlinear elements should be provided.

Line 185 "peak interstorey drifts, IDR,...". Maybe "ratio" is missing here.

Line 234."The beam rotation is not reported because its con-234 tribution is negligible thanks to the beam high stiffness in X direction." Please explain with more detailed information.

In Fig6 (c), it seems the deformation doesn't come to an end. Is it due to the cut of the input wave?

Author Response

Response to Reviewer #1

We would like to thank the anonymous Reviewer for his/her time and effort, as well as for the constructive comments. The new version of the manuscript contains modified parts in the review mode. In this document, you may find a point-by-point discussion. We hope the revised manuscript has satisfactorily accounted for all the comments and now is adequate to be accepted.

Comment

The authors made a great job in analyze the seismic performance of a RC building constructed in Mediterranean area based on non-seismic design codes. Opensees is adopted to handle the FEA and fragility curves for different damage states are provided. The reviewer has following questions for the authors:

How the foundation and soil-structure interaction is modeled/considered in the FE model?

Reply

Thank you for your comment. The following sentence has been added in Section 4 to clarify this point:

“At the ground level, the columns were fully clamped, neglecting the soil-structure interaction”.

Comment

Please explain detailed procedures to acquire/define the values used in zero-length elements simulating nonlinear behavior, as in Fig 3 a, b and c, including My, Mpeak, Mcr, etc..

Reply

Different reinforcement bar configurations characterize the structural elements (beam, column and joint), and their shear strength and/or flexural strength capacity have been evaluated starting from the sectional analysis. Its non-linear behavior is evaluated according to EC- 8 part 3 for beams and columns and according to “Pampanin, S.; Magenes, G.; Carr, A. Modelling of Shear Hinge Mechanism in Poorly Detailed RC Beam-Column Joints. Proc. fib Symp. 2003 Concr. Struct. Seism. Reg. 2003, 126–127” for joints.

In Section 4, the following sentences clarify this aspects:

  • “A first zero-length element with rigid-plastic behavior was used to introduce the elements shear collapse mechanism [37]. The points of the curve were defined according to [3], evaluating the shear strength capacity starting from the sectional analysis as a function of the adopted shear reinforcement.”
  • “The non-linear behavior of the joint was introduced in the model as trilinear behavior calibrated as discussed in [38] on the basis of experimental results in [39][40].”
  • “The moment-rotation law, displayed in Figure 3c, is characterized by different capacity values for each floor, as a function of the axial load, and for the exterior or internal panel joints.”
  • “A second zero-length element was added at the extremity of beams and columns, in series with the shear zero-length element, in order to model the inelastic flexural behavior. The values were obtained starting from the properties of the cross-sections of the different elements and material properties of concrete and steel for reinforcement bars.”

Comment

A figure showing whole FE model, indicated with detailed location of nonlinear elements should be provided.

Reply

Following the Reviewer’s comment, a figure has been added in Section 4.

Comment

Line 185 "peak interstorey drifts, IDR,...". Maybe "ratio" is missing here.

Reply

Thank you for your comment. The word “ratio” has been added.

Comment

Line 234."The beam rotation is not reported because its contribution is negligible thanks to the beam high stiffness in X direction." Please explain with more detailed information.

Reply

Following the Reviewer's comment, the sentence has been modified as follows in order to clarify this isssue:

“The beam rotation is not reported because its contribution is negligible if compared to those of columns and joints. In fact, due to the absence of the capacity design criterion, the building shows a strong beam-weak columns mechanism with beams characterized by almost zero rotations.”

Comment

In Fig6 (c), it seems the deformation doesn't come to an end. Is it due to the cut of the input wave?

Reply

In Figure 6c (Figure 7c in the revised version) only the first 60s of time history analysis are reported. A new sentence has been added in the text to clarify this point:

“The rotation contribution of the structural members converging in the beam-column joint sub-system varies during the dynamic structural analysis, as shown in Figure 7c, where the rotation recorded for the selected joint panel, upper column, and lower column is reported for the first 60 s (out of a total of 120 s) of time-history analysis. After the first 60 s the rotation values are less significant and move toward zero.”

Reviewer 2 Report

Dear authors,

The paper focuses on the evaluating the seismic performance of an existing RC building. In the reviewers' opinion, the paper can be considered for further process. However, the following points need to be clarified (see attachment).

Author Response

Response to Reviewer 2

We would like to thank the anonymous Reviewer for his/her time and effort, as well as for the constructive comments. The new version of the manuscript contains modified parts in the review mode. In this document, you may find a point-by-point discussion. We hope that the revised manuscript has satisfactorily accounted for all the comments and now is adequate to be accepted.

Comment

Dear authors,

The paper focuses on the evaluating the seismic performance of an existing RC building. In the reviewers' opinion, the paper can be considered for further process. However, the following points need to be clarified:

Abstract should be rewritten to describe the main process and the finding as well.

Reply

According to the Reviewer’s comment, the Abstract has been revised to clarify the motivation of the work and its main outcomes as follows:

“In the past decades, the considerable number of worldwide earthquakes caused considerable damage and several building collapses, underlining the high vulnerability of the existing buildings designed without seismic provisions. To this regard, this work analyses the seismic performance of a reinforced concrete building designed without any seismic criteria, characterized by a seismically-stronger and a seismically-weaker direction such as several existing reinforced concrete framed structures designed for vertical load only. The case study building was modelled in OpenSees considering a non-linear three-dimensional model also accounting for the contribution of joint panel deformability on the global behavior. Thirty bidirectional ground motions have been applied to the structure with the highest component alternatively directed along the two principal building directions. The time-history analyses have been performed for eight increasing hazard levels with the aim of evaluating the influence of bidirectional ground motion on structural response and estimating the seismic vulnerability of the building. The seismic performance of the structure are provided in terms of fragility curves for the two principal directions of the building and for different damage states defined according to the EMS-98 scale.”

Comment

English issues:

  • Figures: titles should be reduced to brief, detailed explanations can be provided in the figure or main content.

Reply

Thank for your comment. Where possible, the description of the figures has been moved in the text and the captions have been reduced for almost all the figures.

Comment

  • Line 141: space between "by1D" is needed.

Reply

Done.

Comment

  • Line 141, 142: this sentence should be rewritten for more understanding to readers.

Reply

Following the Reviewer's comment, the sentence has been revised as follows:

"The 3D bare frame model of the structure was realized by employing 1D elastic finite elements for beams and columns, and the non-linear behavior was accounted for by means of flexural and shear springs introduced with zero-length elements, as shown in Figure 3."

Moreover, a figure showing the detailed FE model has been introduced in order to clarify the non-linear modeling.

Comment

  • Math symbols in the text should be in ITALIC format.

Reply

We have corrected them.

Comment

  • Line 286, 287, 288: very long sentence

Reply

According to the Reviewer's comment, the sentence has been summarized as follows:

"The structural analysis results, expressed in terms of IDR, were employed as EDP for building vulnerability assessment through fragility curves."

Comment

  • Line 119,120,121: very long sentence, should be rewritten

Reply

Following the Reviewer's suggestion, the long sentence has been revised.

Comment

  • Section 5: "Structural results" should be change to "Analysis results"

Reply

Thank you for your comment. The title of Section 5 has been modified in "Structural analysis results".

Comment

Ground motion data: a total of 30 GMs are considered, however, the authors did not mention the criteria of selection of ground motions. What can the authors say about this?

Reply

The thirty pairs of time-histories were selected considering a magnitude moment Mw ranging from 6.5 to 8.0; a distance from the hypocentre ranging from 6 to 50 km; a horizontal peak ground acceleration, PGA, varying from 0.07g to 0.48g recorded on site classes B or C according to Eurocode 8. Then, the pairs have been scaled in order to have a geometric mean (geo-mean) of elastic acceleration response spectra spectra-compatible with the elastic spectra of the site.

The following sentence has reported in Section 3:

“The thirty pairs of time-histories were chosen and extracted from the Pacific Earthquake Engineering Research Center (PEER) strong motion database [37] considering: a magnitude moment Mw ranging from 6.5 to 8.0; a distance from the hypocentre ranging from 6 to 50 km; a horizontal peak ground acceleration, PGA, varying from 0.07g to 0.48g recorded on site classes B or C according to Eurocode 8 [3]. The PGA1/PGA2 ratio varies between 0.5 and 2.0, as shown in Table 1.”

Comment

Please explain why the nonlinear approach for modelling is considered.

Reply

Following the Reviewer’s comment, the reasons for adopting a non-linear model have been introduced in the text by adding the statement:

“A non-linear model has been considered in order to capture, for the increasing seismic intensity, the degradation mechanisms of the building until the building collapse”.

Comment

As known, the durability of aged material is affected because of the mechanisms of the external load and environmental incidents, thermal effect, etc. In this study, an existing building constructed in 1960-1970 is evaluated, so what can the authors say about these effects?

Reply

The aged properties of the materials have been introduced in the study by considering in a proper way the long-term effects. In fact, the suitable reduction coefficients suggested by the Eurocode have been used in order to simulate long term phenomena (e.g. shrinkage, creep etc.).

Comment

How does the finite element model verify the existing building?

Reply

The finite element model simulates the seismic behaviour of an archetype representative of the response of several existing reinforced concrete buildings. The building and the structure have been designed and created, thanks to a simulated project of a representative case study, for the seismic analyses but it is a numerical model since the “real building” doesn’t exist.              

Comment

Seismic vulnerability plays an important role in structural design. Several unquoted publications on this effect on structural performance are highly recommended.

  • https://www.sciencedirect.com/science/article/pii/S240584402100935X
  • https://www.tandfonline.com/doi/abs/10.1080/00223131.2020.1724206

Reply

The two references have been added as suggested by the Reviewer.

Round 2

Reviewer 2 Report

Line 188:  There is a mistake in "Structure analysis results".

The authors state in their text that "Structure analysis results". It should be revised to "Structural analysis results"